# GraphDenoiser: An Unsupervised Iterative Framework for Node Label Denoising in Graph-Structured Data

## Abstract

Data annotation errors have always been one of the core challenges in the field of supervised learning: such noise not only interferes with the model's effective learning of the patterns of data distribution, but also directly leads to the model's discriminative bias in target tasks, significantly reducing the predictive accuracy of supervised learning systems. For graph-structured data, due to the uniqueness of the associative relationships between data points, traditional denoising methods struggle to adapt to the noise distribution patterns in this scenario. To address this issue, this paper focuses on the denoising problem of node type labels in graph data and proposes a denoising framework based on unsupervised learning called **GraphDenoiser**. Through multiple rounds of iteration between the node label noise prediction model and the synthetic data generation model, this framework can quantitatively output the noise probability of each node label and accurately locate mislabeled nodes in graph data. This provides a reliable noise diagnosis basis for subsequent label correction and robust graph model training, thereby alleviating the constraints of node mislabeling in graph data on supervised learning performance. Experiments show that under eight different noise injection methods across three datasets, compared with previous methods, the metrics of MCC, and F1 have increased by 22.81%, and 30.51% respectively.

## 1 Introduction

In the era of data-driven artificial intelligence, supervised learning has become a core supporting technology for numerous key applications Goodfellow et al. (2016). However, the performance of supervised learning systems is inherently limited by the quality of training data Recht et al. (2019); Yang et al. (2023). As a prevalent and intractable challenge in this field, label noise not only distorts the model's learning process of data distribution patterns but also causes biases in model predictions, ultimately reducing the reliability of downstream tasks Li & Vasconcelos (2019). This problem is even more prominent in the scenario of graph-structured data Jin et al. (2020): as a crucial data type for depicting relational connections between entities, graph data has become increasingly important in both academic and industrial fields in recent years. Unlike independent and identically distributed tabular data, graph data exhibits inherent inter-node dependency — node labels are determined not only by their own features but also closely related to the attributes and labels of neighboring nodes Xu et al. (2018); Cheng et al. (2024). This structural correlation leads to the propagation of label noise in the graph, making traditional denoising methods unable to adapt to the non-independent noise distribution of graph data and ultimately rendering them ineffective Dong & Kluger (2023).

The existing graph label denoising methods have the following limitations, which constitute the core motivation of this study:

1. **Neglect of graph structural correlation**: Traditional denoising frameworks focus on independent samples and fail to leverage the relational information of graphs Arazo et al. (2019b); Zhu et al. (2022). These methods treat each node as an isolated instance, ignoring the key characteristic that the credibility of a node's label is strongly correlated with the labels and features of its neighbors.

2. **Over-reliance on clean annotations**: Most graph denoising methods require a large number of manually verified clean labels to train noise detectors, which is difficult to implement in real-

world scenarios — high-quality annotations are often cost-prohibitive or even unavailable Han et al. (2018); Xia et al. (2023).

To address the above issues, this paper proposes an unsupervised graph label denoising framework called **GraphDenoiser**. Through multi-round iterative optimization of a node label noise prediction model and a simulated noise generation model, the framework achieves accurate detection of label noise in graphs. Without the need for additional clean annotations, the framework fully utilizes the structural information of the graph and self-generated noise data to train a robust noise detection model.

The main contributions of this paper are as follows:

1. **Proposing an iteratively optimized unsupervised denoising framework**: For the first time, forward graph noise generation and reverse graph noise prediction are closely integrated, and the two are optimized iteratively. This not only improves the quality of data generation but also enhances the prediction accuracy. The entire process does not rely on additional clean annotations, significantly reducing the application cost of the method in practical scenarios.

2. **Designing a noise simulation mechanism in the embedding space**: A pre-trained autoencoder is used to achieve controllable and semantically consistent noise generation, overcoming the limitations of adding noise in the original feature space and improving the diversity of training data for the noise predictor.

3. **Proposing an experience replay-enhanced iterative optimization strategy**: An experience replay mechanism is introduced to store noise feature-label sample pairs generated in multiple iterations. During model training, the model no longer relies solely on the data of the current round but randomly samples from the historical sample pool. This strategy prevents the model from overfitting to transient noise patterns, reduces training fluctuations, and significantly enhances the training stability of the noise prediction model and its robustness to different noise distributions.

4. **Empirical performance advantages**: Under 8 noise configurations across 3 graph datasets, the MCC, and F1 scores of the proposed method are improved by 22.81%, and 30.51% respectively compared with existing methods, verifying its effectiveness.

## 2 RELATED WORK

### 2.1 GRAPH NEURAL NETWORKS

GNNs (Graph Neural Networks) are one of the most effective methods for graph learning tasks. Early GNN methods Scarselli et al. (2008) iteratively update node states through recurrent units and enables end-to-end learning of graph structures. GCN (Graph Convolutional Network) Kipf (2016) extends convolutional operations to graph structures and achieves feature aggregation by weighted summation of local neighbor features, greatly improving the performance and training stability of node classification tasks. GAT (Graph Attention Network) Veličković et al. (2017) introduces a self-attention mechanism to dynamically learn the attention coefficients of neighbor nodes and performs better in heterogeneous neighbor scenarios. GraphSAGE Hamilton et al. (2017) reduces computational complexity by randomly sampling neighbor nodes, enables efficient training of graphs with millions of nodes, and promotes the application of GNNs in industrial scenarios.

However, erroneous labels in graph datasets undermine the authenticity of graph data, interfere with the common feature aggregation and structure learning processes of graph neural networks, and cause the model to learn patterns that deviate from reality. This not only reduces task accuracy but also affects the reliability of its application in industrial scenarios, becoming a common problem that restricts the performance of graph neural networks Fox & Rajamanickam (2019); Dong & Kluger (2023).

### 2.2 GNN TRAINING UNDER NOISY DATA

Graph data noise is widely present in node features, topological structures, and label information. It directly undermines the ability of Graph Neural Networks to learn the structural correlations of graphs, leading to embedding distortion and degraded performance in downstream tasks. Graph

denoising techniques are mainly categorized into two types: noise localization and robust training NT et al. (2019); Dai et al. (2021); Zhu et al. (2021); Qian et al. (2023). These two categories work in synergy to form a complete noise-resistant framework. Specifically, noise localization can be further divided into unsupervised graph noise localization methods Chen & Eldar (2021); Zhang et al. (2022); Yang et al. (2024) and supervised graph noise localization methods Xia et al. (2023) .

Currently, graph denoising techniques face three core limitations:

1. Supervised localization relies on annotations and exhibits poor generalization in scenarios where noisy annotations are scarce.

2. The noise judgment thresholds of unsupervised methods are mostly empirically set, which easily leads to misjudgment for complex graph structures.

3. Robust training improves noise resistance through additional regularization, but this significantly increases computational overhead, making it difficult to balance robustness and efficiency.

### 2.3 GRAPH NOISE GENERATION MODEL

In this paper, the synthetic mislabeled dataset generation mechanism proposed in the GraphCleaner Li et al. (2023) is adopted as the graph noise generation model, which specifically consists of two core stages:

1. **Mislabeled Transition Matrix Estimation**: Using the prediction information of the pre-trained graph neural network, the mislabel transition matrix $\hat{Q}$ is learned on the validation set. The process is as follows: First, the high-confidence predictions of the base classifier are used to approximate the ground-truth labels, and the joint distribution of "**observed labels - high-confidence predicted labels**" is counted. Subsequently, the joint distribution is converted into a conditional probability distribution via Bayes' theorem, and finally the matrix $\hat{Q}$ is obtained, realizing the modeling of class-dependent mislabeling patterns.

2. **Noisy Graph Construction**: For the sampled candidate node set $V$ for which mislabeled data needs to be generated, controllable label flipping is performed on it based on $\hat{Q}$ to generate the noisy label matrix $Y_c$. Then, combined with the adjacency matrix $A$ of the original graph, the graph $G_c$ with synthetic noise and the features $X_i$ of each node $i$ based on k-hop neighborhood information are obtained.

The advantage of this mechanism lies in that the noise distribution is close to real-world scenarios, which avoids the limitations of random label flipping and provides a possibility for sampling to obtain high-quality noisy data.

## 3 METHODOLOGY

### 3.1 OVERVIEW

In this paper, an iterative unsupervised graph data noise detection framework is proposed. It obtains supervised data for the node label noise prediction model through data sampling, and uses this data to acquire realistic input data that conforms to the noise labels of the supervised data via the graph noise generation model. This data is then used to train the node label noise prediction model, which continuously improves the prediction accuracy and data sampling quality during the iterative process, thereby achieving good noise node prediction. The overall framework is shown in Figure 1.

### 3.2 AUTOENCODER

To ensure that the graph information sampled in the subsequent sampling stage still contains the implicit relationships in the real graph data, the autoencoder is first pre-trained. The autoencoder consists of an encoder and a decoder: the encoder maps the input features to an embedding space, while the decoder recovers the original features from this embedding space.

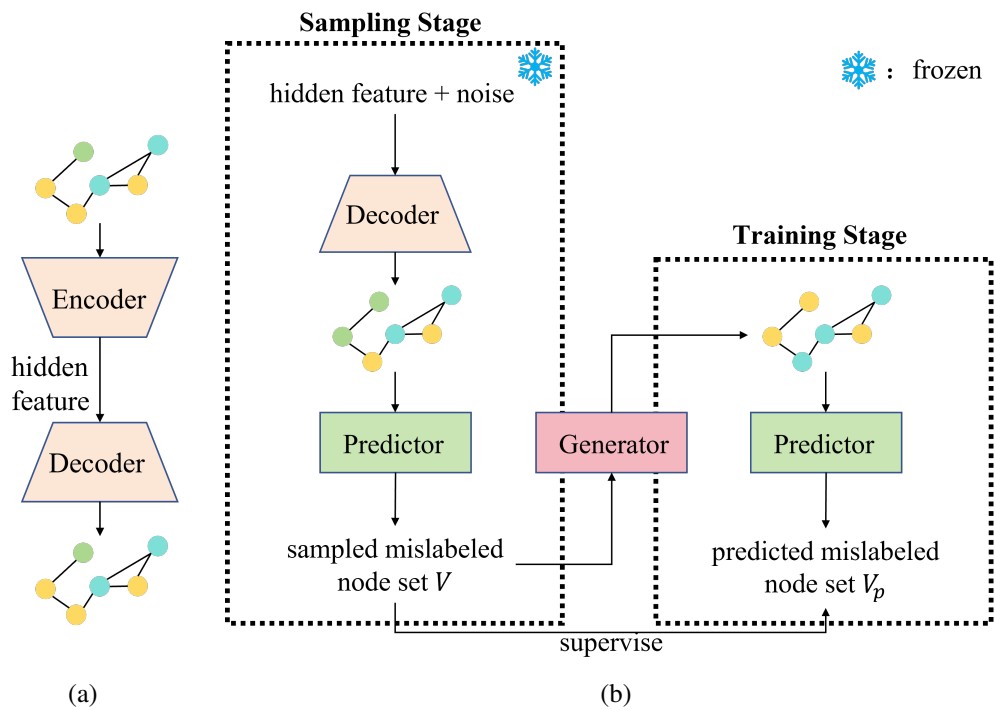

Figure 1: The Framework Diagram of GraphDenoiser. Wherein, (a) illustrates the pre-training process of the AutoEncoder, and (b) illustrates the single-iteration training process of GraphDenoiser. The Decoder used in (b) is the same as the corresponding pre-trained module in (a), and the Predictor for the Sampling Stage comes from the Predictor trained in the Training Stage of the previous iteration.The Generator refers to the graph noise generation model mentioned in the previous section.

The pre-training aims to minimize the MSE (Mean Squared Error) between the input and the reconstructed features, enabling the autoencoder to learn an accurate feature reconstruction mapping. For the input graph feature matrix $X$, the loss function of the autoencoder is defined as follows:

$$L_{AE} = \frac{1}{n} \sum_{i=1}^{n} \|X_i - \text{Decoder}(\text{Encoder}(X_i))\|_2^2$$

where $n$ denotes the number of samples, and Encoder($\cdot$) and Decoder($\cdot$) represent the forward propagation operations of the encoder and decoder, respectively. After pre-training, the encoder constrains the embeddings to the interval [0, 1] through Sigmoid activation, providing a stable feature space for subsequent noise injection.

### 3.3 SAMPLING STAGE

In this stage, we add Gaussian noise to the hidden features corresponding to the real data, and use the decoder from the pre-trained autoencoder to obtain sampled graph data similar to the real data. For these data, we use the frozen node label noise prediction model to predict the probability $p_i$ that each node $i$ is a mislabeled node, and then obtain a set of mislabeled nodes $V$ that is similar to the mislabeled scenario in the real graph data, where:

$$V = \{i \mid p_i > (1 - p_i)\} = \{i \mid p_i > 0.5\}$$

It is worth emphasizing that, compared with traditional mislabeled node identification schemes, the above-mentioned classification method has significant advantages: traditional methods often

require manual adjustment of various thresholds, and slight changes in these thresholds may lead to substantial fluctuations in identification results. In contrast, the interpretable fixed threshold of this method (i.e., the probability that a node label is noise is greater than the probability that it is not) not only simplifies the operation process, but also reduces the uncertainty caused by human intervention, thereby greatly enhancing the adaptability and robustness of the model in scenarios involving different types of graph data.

However, due to the prediction biases of the node label noise prediction model, the sampled graph data and the set of mislabeled nodes cannot be directly used as the input and output for supervised training. For the set of mislabeled nodes, we use the graph noise generation model to generate the corresponding noisy graph $G_c$. $G_c$ and $V$ thus form a set of data that can be used to supervisedly train the node label noise prediction model.

## 3.4 TRAINING STAGE

To enable the model to simultaneously learn a broader range of data features and alleviate the training instability caused by dynamic changes in data distribution during the iterative process, we introduce an experience replay mechanism at this stage. The "graph feature - noisy label" sample pairs sampled in the Sampling Stage of each iteration are stored in a sliding window experience replay buffer. The buffer adopts a first-in-first-out strategy: when new samples are added and the total number exceeds the preset capacity, the earliest stored samples are removed.

This design not only ensures the timeliness of samples but also maintains the diversity and representativeness of data distribution by retaining samples from different iteration stages. During each Training Stage, a batch of samples is randomly extracted from this buffer to update the model parameters. This allows the model to learn the discriminative patterns between noisy samples and clean samples from more abundant historical data.

In the node label noise prediction task, the core objective is to accurately identify nodes with label biases in the dataset. The model architecture designed to achieve this goal exhibits a high degree of flexibility — theoretically, any model that can effectively capture node feature information and support binary classification tasks can serve as the basic architecture for a node label noise prediction model. In this paper, we employ an MLP (Multi-Layer Perceptron) as the core model.

For the design of the model's loss function, we adopt the BCE (Binary Cross-Entropy) Loss, whose specific formula is:

$$L = -\frac{1}{N} \sum_{i=1}^{N} [y_i \log(p_i) + (1 - y_i) \log(1 - p_i)]$$

where $N$ represents the number of samples, $y_i$ denotes the true label of the i-th sample, $p_i$ stands for the probability predicted by the model that the i-th sample belongs to the noisy label category.

## 3.5 OVERALL ITERATIVE PROCESS

This paper takes "**noisy data sampling → training of the node label noise prediction model**" as the core cycle. With the support of feature expression from the autoencoder and continuous interaction between the two stages, the framework achieves the gradual iterative improvement of noise detection performance. Finally, the trained model is used to perform inference on the test set, output the noise probability of samples, and complete the graph data noise detection.

Training the node label noise prediction model can improve its prediction accuracy, thereby enabling the data sampled in the Sampling Stage to be closer to the real data. At the same time, as the data sampled in the Sampling Stage keeps approaching the real data, the data quality is continuously improved, which in turn enhances the training quality of the node label noise prediction model in the Training Stage. The two stages continuously iterate and optimize each other. The pseudocode of the iterative training process is shown in Algorithm 1.

---

**Algorithm 1** Iterative Training Process of GraphDenoiser

---

**Require:** features of each node in the graph based on k-hop neighborhood information $X$,
    encoder and decoder from the pre-trained autoencoder $Encoder$ and $Decoder$ ,
    graph noise generation model $G$,
    indicator function $\mathbb{I}(\cdot)$ ,
    Hyperparameters $\{\text{lr}, \text{wd}, \text{batch\_size}, \text{buffer\_cap}, \dots\}$
  1: Initialize node label noise prediction model $P$
  2: Initialize sliding window buffer $\mathcal{B}$ with capacity buffer\_cap
  3: **for** iter $= 1$ **to** max\_iter **do**
  4:    $z = Encoder(X)$
  5:    $z_{sampled} = z + \mathcal{N}(0, \sigma^2)$   (clamped to $[0, 1]$)
  6:    $X_{sampled} = Decoder(z_{sampled})$
  7:    freeze parameters in $P$
  8:    $\mathbf{y}_{NewData} = \mathbb{I}(P(\mathbf{X}_{sampled}) > 0.5)$
  9:    unfreeze parameters in $P$
10:    $X_{NewData} = G(y_{NewData})$
11:    store $(X_{NewData}, y_{NewData})$ in $\mathcal{B}$
12:    **if** size of $\mathcal{B}$ is greater than buffer\_cap **then**
13:        Remove oldest data
14:    **end if**
15:    **if** $|\mathcal{B}| \geq$ batch\_size **then**
16:        Sample batch\_size data $(X_{batch}, y_{batch})$ from $\mathcal{B}$
17:        $y_{predicted} = \mathbb{I}(P(X_{batch}) > 0.5)$
18:        Compute loss between $y_{predicted}$ and $y_{batch}$
19:        Update parameters of model $M$ via backpropagation
20:    **end if**
21: **end for**

---

## 4 EXPERIMENT

### 4.1 DATASET DESIGN

In the experimental validation phase of this study, to ensure the generality, reliability, and practical reference value of the experimental results, we selected three classic benchmark graph datasets, namely Cora, CiteSeer, and PubMed, as the basis for raw data. These three datasets have become standard test sets for evaluating the performance of graph learning algorithms due to their moderate data scale, clear topological structure, and coverage of features from different fields. Their specific introductions are as follows Yang et al. (2016):

**Cora**: It focuses on the academic paper citation network in the field of machine learning, containing 2708 paper nodes. Each paper corresponds to one research topic (with a total of 7 categories). The edges between nodes represent the citation relationships among papers, and the node features are 1433-dimensional vectors.

**CiteSeer**: It is an academic paper citation network covering 3327 paper nodes in the field of computer science (with a total of 6 categories), and the node features are 3703-dimensional vectors.

**PubMed**: It is a literature citation network oriented to the field of biomedicine. With a relatively larger scale, it includes 19717 biomedical literature nodes (with a total of 3 categories). The node features are 500-dimensional vectors, and its network topological structure is more complex, which can effectively test the adaptability of algorithms to large-scale graph data.

To simulate the problem of node label mislabeling in real-world scenarios, which is prone to be caused by factors such as data collection errors and manual annotation biases - we designed a two-dimensional combination scheme of "**noise injection method × noise frequency**" for each of the aforementioned raw datasets, and constructed multiple groups of graph datasets with node type mislabeling. The specific design logic and implementation details are as follows:

Table 1: Results of Comparative Experiments on 8 Noise Injection Methods Across 3 Datasets. 'sym' and 'asym' denote symmetric and asymmetric noise. Items in bold represent the best results, while underlined items represent the second-best results.

| Noise Type | Noise Rate | Method | Cora | | | CiteSeer | | | PubMed | | |
|---|---|---|---|---|---|---|---|---|---|---|---|
| | | | P@T | MCC | F1 | P@T | MCC | F1 | P@T | MCC | F1 |
| sym | 10% | baseline | 0.175 | 0.388 | 0.296 | 0.235 | 0.333 | 0.336 | 0.444 | 0.478 | 0.515 |
| | | DYB | 0.289 | 0.437 | 0.425 | 0.184 | 0.320 | 0.333 | 0.283 | 0.249 | 0.281 |
| | | AUM | 0.598 | 0.038 | 0.179 | 0.439 | 0.021 | 0.179 | **0.677** | 0.037 | 0.182 |
| | | CL | 0.722 | 0.608 | 0.633 | 0.520 | 0.432 | 0.448 | 0.556 | 0.435 | 0.474 |
| | | GraphCleaner | 0.794 | 0.766 | 0.785 | 0.531 | 0.489 | 0.514 | 0.535 | 0.496 | 0.547 |
| | | Ours | **0.804** | **0.795** | **0.809** | **0.541** | **0.560** | **0.596** | 0.546 | **0.547** | **0.593** |
| sym | 7.5% | baseline | 0.086 | 0.259 | 0.156 | 0.153 | 0.256 | 0.237 | 0.452 | 0.445 | 0.482 |
| | | DYB | 0.286 | 0.405 | 0.371 | 0.125 | 0.297 | 0.274 | 0.137 | 0.000 | 0.136 |
| | | AUM | 0.600 | 0.019 | 0.132 | 0.458 | 0.034 | 0.136 | **0.616** | 0.031 | 0.138 |
| | | CL | 0.629 | 0.544 | 0.564 | 0.486 | 0.383 | 0.372 | 0.466 | 0.379 | 0.393 |
| | | GraphCleaner | 0.686 | 0.668 | 0.682 | 0.472 | 0.501 | 0.494 | 0.452 | 0.413 | 0.449 |
| | | Ours | **0.743** | **0.700** | **0.708** | **0.500** | **0.553** | **0.560** | 0.466 | **0.439** | **0.479** |
| sym | 5% | baseline | 0.106 | 0.289 | 0.189 | 0.208 | 0.293 | 0.290 | 0.449 | 0.408 | 0.438 |
| | | DYB | 0.277 | 0.350 | 0.286 | 0.063 | 0.239 | 0.198 | 0.184 | 0.148 | 0.141 |
| | | AUM | 0.596 | 0.028 | 0.091 | 0.354 | 0.021 | 0.092 | **0.571** | 0.029 | 0.095 |
| | | CL | 0.660 | 0.558 | 0.555 | **0.458** | 0.368 | 0.318 | 0.429 | 0.343 | 0.318 |
| | | GraphCleaner | **0.723** | 0.648 | 0.637 | 0.375 | 0.398 | 0.378 | 0.449 | 0.449 | 0.439 |
| | | Ours | 0.702 | **0.743** | **0.755** | **0.458** | **0.510** | **0.524** | 0.490 | **0.481** | **0.505** |
| sym | 2.5% | baseline | 0.095 | 0.212 | 0.160 | 0.318 | 0.349 | 0.359 | 0.500 | 0.402 | 0.400 |
| | | DYB | 0.143 | 0.222 | 0.130 | 0.000 | 0.145 | 0.088 | 0.208 | 0.149 | 0.087 |
| | | AUM | 0.476 | 0.012 | 0.041 | 0.182 | 0.017 | 0.044 | **0.625** | 0.017 | 0.047 |
| | | CL | 0.476 | 0.335 | 0.320 | 0.318 | 0.197 | 0.138 | 0.500 | 0.304 | 0.219 |
| | | GraphCleaner | 0.524 | 0.506 | 0.462 | 0.273 | 0.303 | 0.232 | 0.542 | 0.379 | 0.309 |
| | | Ours | **0.619** | **0.629** | **0.636** | **0.364** | **0.395** | **0.378** | 0.583 | **0.617** | **0.595** |
| asym | 10% | baseline | 0.103 | 0.307 | 0.187 | 0.102 | 0.158 | 0.161 | 0.283 | 0.307 | 0.350 |
| | | DYB | 0.309 | 0.407 | 0.417 | 0.174 | 0.268 | 0.299 | 0.152 | 0.000 | 0.180 |
| | | AUM | 0.536 | 0.040 | 0.179 | 0.378 | 0.030 | 0.180 | **0.667** | 0.040 | 0.183 |
| | | CL | 0.608 | 0.598 | 0.624 | 0.357 | 0.375 | 0.409 | 0.465 | 0.416 | 0.456 |
| | | GraphCleaner | 0.629 | 0.616 | 0.650 | **0.429** | 0.420 | 0.463 | 0.495 | 0.452 | 0.509 |
| | | Ours | **0.670** | **0.652** | **0.686** | 0.408 | **0.435** | **0.483** | 0.505 | **0.474** | **0.523** |
| asym | 7.5% | baseline | 0.100 | 0.234 | 0.173 | 0.056 | 0.111 | 0.095 | 0.288 | 0.290 | 0.330 |
| | | DYB | 0.257 | 0.347 | 0.311 | 0.111 | 0.225 | 0.229 | 0.178 | 0.202 | 0.212 |
| | | AUM | 0.571 | 0.040 | 0.134 | 0.361 | 0.027 | 0.136 | **0.650** | 0.036 | 0.138 |
| | | CL | 0.529 | 0.509 | 0.513 | **0.444** | 0.357 | 0.354 | 0.356 | 0.370 | 0.385 |
| | | GraphCleaner | 0.614 | 0.586 | 0.602 | 0.403 | 0.389 | 0.403 | 0.411 | 0.388 | 0.434 |
| | | Ours | **0.657** | **0.641** | **0.667** | 0.417 | **0.412** | **0.449** | 0.452 | **0.445** | **0.477** |
| asym | 5% | baseline | 0.043 | 0.118 | 0.077 | 0.167 | 0.270 | 0.250 | 0.245 | 0.209 | 0.247 |
| | | DYB | 0.213 | 0.315 | 0.252 | 0.063 | 0.218 | 0.182 | 0.143 | 0.175 | 0.154 |
| | | AUM | 0.447 | 0.021 | 0.091 | **0.354** | 0.030 | 0.093 | **0.531** | 0.028 | 0.095 |
| | | CL | 0.553 | 0.518 | 0.510 | 0.417 | 0.329 | 0.291 | 0.286 | 0.303 | 0.289 |
| | | GraphCleaner | 0.619 | 0.585 | 0.525 | 0.333 | 0.374 | 0.360 | 0.408 | 0.358 | 0.370 |
| | | Ours | **0.638** | **0.628** | **0.633** | **0.354** | **0.418** | **0.400** | 0.388 | **0.393** | **0.405** |
| asym | 2.5% | baseline | 0.048 | 0.120 | 0.083 | 0.091 | 0.090 | 0.105 | 0.250 | 0.253 | 0.265 |
| | | DYB | 0.143 | 0.231 | 0.137 | 0.000 | 0.128 | 0.082 | 0.167 | 0.153 | 0.093 |
| | | AUM | 0.429 | 0.015 | 0.042 | **0.227** | 0.013 | 0.043 | **0.542** | 0.021 | 0.048 |
| | | CL | **0.571** | 0.454 | 0.410 | 0.182 | 0.203 | 0.143 | 0.250 | 0.264 | 0.193 |
| | | GraphCleaner | 0.476 | 0.596 | 0.539 | 0.182 | 0.231 | 0.197 | 0.333 | 0.361 | 0.296 |
| | | Ours | 0.524 | **0.627** | **0.618** | **0.227** | **0.236** | **0.237** | 0.375 | **0.369** | **0.385** |

1. **Design of Noise Injection Methods**: Considering the different generation mechanisms of label noise in reality, we adopted two representative noise injection strategies, symmetric and asymmetric, to cover different types of label error scenarios Tan et al. (2021); Xia et al. (2022); Chen et al. (2019).

2. **Setting of Noise Frequency**: With reference to the statistical results of label noise frequency in real graph datasets from existing literature, we selected four gradient noise frequencies: 10%, 7.5%, 5%, and 2.5%, so as to cover real-world scenarios ranging from high noise to low noise Northcutt et al. (2021a).

3. **Dataset Construction**: Through the full combination strategy of "**noise injection method × noise frequency**", we constructed 8 groups of graph datasets with node type mislabeling for each

Table 2: Results of Ablation Studies on 8 Noise Injection Methods for the Cora Dataset. 'sym' and 'asym' denote symmetric and asymmetric noise. w/o Buffer denotes removing the sliding window experience replay buffer. w/o AE denotes further removing the autoencoder based on the former. Items in bold indicate the best results.

| | | sym | | | | | | | | | | |
|---|---|---|---|---|---|---|---|---|---|---|---|---|
| Method | | 10% | | | 7.5% | | | 5% | | | 2.5% | |
| | P@T | MCC | F1 | P@T | MCC | F1 | P@T | MCC | F1 | P@T | MCC | F1 |
| Ours | 0.804 | **0.795** | **0.809** | 0.743 | **0.700** | **0.708** | 0.702 | **0.743** | **0.755** | **0.619** | **0.629** | **0.636** |
| w/o Buffer | **0.814** | 0.473 | 0.461 | 0.714 | 0.403 | 0.369 | 0.723 | 0.327 | 0.259 | 0.571 | 0.562 | 0.571 |
| w/o AE | 0.794 | 0.404 | 0.393 | **0.757** | 0.331 | 0.293 | **0.745** | 0.264 | 0.204 | **0.619** | 0.192 | 0.108 |
| | | asym | | | | | | | | | | |
| Method | | 10% | | | 7.5% | | | 5% | | | 2.5% | |
| | P@T | MCC | F1 | P@T | MCC | F1 | P@T | MCC | F1 | P@T | MCC | F1 |
| Ours | **0.670** | **0.652** | **0.686** | **0.657** | **0.641** | **0.667** | **0.638** | **0.628** | **0.633** | 0.524 | **0.627** | **0.618** |
| w/o Buffer | 0.660 | 0.646 | 0.679 | 0.629 | 0.614 | 0.630 | 0.617 | 0.361 | 0.297 | 0.429 | 0.611 | 0.590 |
| w/o AE | 0.650 | 0.384 | 0.377 | 0.614 | 0.318 | 0.285 | 0.575 | 0.273 | 0.215 | **0.619** | 0.157 | 0.086 |

raw dataset (calculated as 2 noise injection methods × 4 noise frequencies). Finally, a total of 24 groups of experimental graph datasets with node mislabeling were constructed.

## 4.2 METRICS DESIGN

This paper adopts three metrics — P@T, MCC, and F1 — as the evaluation metrics for the model. These three types of metrics complement each other from distinct evaluation perspectives: P@T ensures the accuracy of high-priority results, MCC addresses the evaluation bias in class imbalance scenarios, and F1 balances the trade-off between precision and recall. The combination of the three can comprehensively cover the model's performance across different scenarios, providing a reliable basis for the objective assessment of model effectiveness Chicco & Jurman (2020); Diallo et al. (2024).

## 4.3 COMPARATIVE EXPERIMENTS

We conducted comparative experiments among GraphDenoiser, DYB Arazo et al. (2019a), AUM Pleiss et al. (2020), CL Northcutt et al. (2021b), GraphCleaner Li et al. (2023) and the baseline. Here, the baseline simply treats samples whose argmax predictions differ from the given labels as mislabeled nodes. Experiments were performed on the 24 noisy graph datasets mentioned above, and the results are presented in Table 1.

Analysis of the experimental results shows that our method performs the best in most metrics. To more intuitively demonstrate the performance improvement of GraphDenoiser, this study conducts a quantitative comparison between GraphDenoiser and the best-performing method among previous methods on each dataset (i.e., for each dataset, the method with the optimal comprehensive performance is selected fromDYB, AUM, CL, GraphCleaner and the baseline). The results indicate that in terms of the average performance across 24 noisy graph datasets, the MCC metric of GraphDenoiser is 22.81% higher than that of the previous best method, and the F1 score is 30.51% higher than that of the previous best method. This significant improvement shows that GraphDenoiser can effectively overcome the performance bottlenecks of traditional methods under complex noise distributions and diverse graph structures, and greatly improve the recognition accuracy of mislabeled nodes.

## 4.4 ABLATION STUDIES

To verify the effectiveness of each module in our method, we constructed eight noisy graph datasets based on the Cora dataset. We sequentially removed the **sliding window experience replay buffer**

and **autoencoder** from the complete method, followed by testing its performance. The experimental results are presented in Table 2.

Analysis of the experimental results reveals that all metrics decreased significantly after removing the sliding window experience replay buffer, indicating that this module contributes to the effectiveness of the proposed method. On this basis, after further removing the autoencoder, it was found that although P@T showed no significant change, MCC and F1 decreased sharply. This proves that the autoencoder also plays a crucial role in ensuring the effectiveness of the proposed method.

In conclusion, all modules of the proposed method support each other functionally and are indispensable. Together, they form a key guarantee for the model to achieve high-performance detection of node label mislabeling in noisy data scenarios.

## 5 CONCLUSION

Addressing the core issues where node label noise in graph-structured data interferes with the performance of supervised learning and traditional denoising methods struggle to adapt to the non-independent distribution characteristics of graph data, this study proposes an iterative graph label denoising framework called GraphDenoiser based on unsupervised learning. This framework systematically addresses the key limitations of existing methods, such as neglecting graph structural correlations and over-relying on clean annotations, and provides an efficient and low-cost solution for detecting mislabeled nodes in graph data.

The experimental results fully verify the effectiveness and superiority of the proposed method: Under 8 noise configurations across 3 graph datasets, GraphDenoiser achieves outstanding performance in the three core metrics of P@T, MCC, and F1. Compared with existing methods, the average improvements in MCC and F1 reach 22.81% and 30.51% respectively. Further ablation studies show that each module supports each other functionally and is indispensable, collectively forming a key guarantee for the model's robustness.

## 6 PROSPECT

Future research can further expand the application scenarios of the framework. For instance, it can be applied to label denoising tasks for heterogeneous graphs and dynamic graphs, or focus on denoising the edges between nodes and edge labels. Ultimately, this aims to achieve comprehensive mislabel detection for various types of graphs.

## 7 THE USE OF LARGE LANGUAGE MODELS (LLMS)

In this paper, LLMs are used to detect spelling errors and grammatical errors.

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
