# OpenReview forum: "GraphDenoiser: An Unsupervised Iterative Framework for Node Label Denoising in Graph-Structured Data"
_ICLR.cc/2026/Conference — ICLR 2026 Conference Withdrawn Submission_

### Official Review · Reviewer_LrUu · 2025-10-27

**Soundness:** 3
**Presentation:** 3
**Contribution:** 2
**Rating:** 4
**Confidence:** 3

**Summary:**

This paper addresses the issue of correcting data annotation errors within graph-structured data. Unlike regular data, annotation errors on graph data can propagate to neighboring nodes through message passing, making the correction process significantly more complex. To tackle this, the authors propose pre-training a model on synthetic annotation errors and subsequently applying the pre-trained model to rectify errors in the target dataset.

**Strengths:**

* The paper explores the relatively under-researched problem of annotation error correction in graph-structured data, tackling an important and challenging issue.
* Experimental results demonstrate the effectiveness of the proposed method.

**Weaknesses:**

* The paper does not clearly highlight the improvements of GraphDenoiser over GraphCleaner, a related method that also utilizes pre-training with synthetic annotation errors. Clarifying the contribution and advancements of GraphDenoiser would strengthen the paper.
* The method is only validated on synthetic datasets. Incorporating evaluations on real-world datasets with annotation errors would enhance the practical applicability and robustness of the proposed approach.
* There are inconsistencies in notation, such as the use of y_{NewData} in Algorithm 1 (lines 8 and 10). Ensuring consistent symbols throughout the paper would improve readability and clarity.

**Questions:**

* In alogorithm 1, line 10, should the input to G also include X?
* Does the noise injection process used during the pre-training stage and test stage follow the same methodology?

---

### Official Review · Reviewer_M3Ui · 2025-10-27

**Soundness:** 2
**Presentation:** 2
**Contribution:** 2
**Rating:** 2
**Confidence:** 5

**Summary:**

This paper proposes GraphDenoiser, an unsupervised iterative framework for denoising node labels in graph-structured data. The framework jointly trains a node label noise prediction model and a graph noise generation model in an iterative manner, supported by a pre-trained autoencoder for stable feature embedding and an experience replay mechanism to improve training robustness. Experiments are conducted on three standard datasets (Cora, CiteSeer, PubMed) under multiple synthetic noise settings, showing performance improvements over existing baselines such as GraphCleaner, CL, and DYB.

**Strengths:**

1. The proposed iterative process combining noise generation and noise prediction is conceptually interesting .

2. The paper is clearly written, with a structured methodology and detailed experimental results.

**Weaknesses:**

1-The paper is primarily heuristic. No theoretical justification or convergence analysis is provided for the iterative denoising process. It remains unclear why or under what conditions the iterative scheme improves noise detection.

2-The key ideas (autoencoder pretraining, experience replay, synthetic noise generation, and iterative training) are borrowed from well-known concepts in fields like generative modeling, denoise included. Their combination for graph denoising is incremental rather than fundamentally novel.

3- Some strong recent baselines for graph robustness or denoising are either not compared or insufficiently discussed.

4- The method involves multiple components (autoencoder, noise generator, replay buffer) but lacks clear intuition about each part’s contribution beyond empirical observation. The ablation study confirms importance but not mechanism.

5-The introduction and methodology sections are verbose and read more like a technical report than a scientific argument.

6-It inherently a threshold method which need set a gap threshold, eg P> 0.5

**Questions:**

Why is the first-in-first-out strategy adopted instead of alternative approaches?

---

### Official Review · Reviewer_mpV9 · 2025-10-29

**Soundness:** 1
**Presentation:** 1
**Contribution:** 1
**Rating:** 2
**Confidence:** 4

**Summary:**

The paper proposes GraphDenoiser, an unsupervised, iterative framework to detect mislabeled nodes in graph data. The loop alternates between (i) a node-label noise predictor and (ii) a graph noise generator adapted from GraphCleaner that estimates a label-transition matrix and flips labels accordingly, producing synthetic noisy graphs used to train the predictor. An autoencoder injects Gaussian noise in embedding space for sampling, and an experience-replay buffer stabilizes training. Experiments on Cora, CiteSeer, and PubMed with synthetic symmetric/asymmetric noise at several rates report improvements in MCC and F1 over baselines.

**Strengths:**

- The paper adopts several sensible metrics (P@T, MCC, F1) and conducts experiments with a comprehensive synthetic noise grid across datasets

- Experimental results show decent empirical gains of the proposed method

**Weaknesses:**

- The presentation is poor, due to numerous missing logical connections and insufficient explanation of key steps, leaving many implementation and reasoning details unclear.

- The claimed limitations have already been addressed by recent works [1–4], which consider graph structure and reduce dependence on clean labels, making the paper’s motivation and contribution unconvincing.

- Limited method novelty: mostly a recombination of established components (transition-matrix noise simulation, confident/noise-aware training, AE perturbation, replay). The “first-time” claim is unsubstantiated.

- “Unsupervised” is overstated: the generator estimates a mislabel transition matrix using a pretrained GNN and validation data, which contradicts the unsupervised framing and may entangle the evaluation with the same supervision used to synthesize labels.

- Narrow evaluation: only classic citation graphs with synthetic noise; no real noisy-label benchmarks, heterophilous graphs, or larger modern datasets to test scalability and external validity.

- Methodological under-specification: fixed 0.5 threshold for noise decisions is asserted to be “interpretable” but lacks calibration analysis. Sensitivity to AE architecture, σ, and replay capacity is not explored.

- The references and baselines are outdated, with the recent baselines only from 2023, ignoring relevant advances from 2024–2025.

[1] Dong, M., & Kluger, Y. (2023, July). Towards understanding and reducing graph structural noise for GNNs. In International Conference on Machine Learning (pp. 8202-8226). PMLR.

[2] Wu, J., Hu, R., Li, D., Huang, Z., Ren, L., & Zang, Y. (2024). Robust heterophilic graph learning against label noise for anomaly detection. Structure, 4(v5), v6.

[3] Li, K., Sun, J., Lou, J., Feng, Z., Zhou, H., Wu, C., ... & Li, J. Leveraging Peer-Informed Label Consistency for Robust Graph Neural Networks with Noisy Labels.

[4] Wang, Z., Sun, D., Zhou, S., Wang, H., Fan, J., Huang, L., & Bu, J. (2024). Noisygl: A comprehensive benchmark for graph neural networks under label noise. Advances in Neural Information Processing Systems, 37, 38142-38170.

**Questions:**

1. Why is the approach unsupervised if the generator requires a pretrained GNN and a validation set?

2. Why is 0.5 appropriate for all datasets/noise regimes? Did you try calibrating the predictor (e.g., temperature scaling) or optimizing thresholds on held-out data?

3. How does the method handle heterophily and structural noise (edge flips/additions) rather than label noise alone? Any results on heterophilous graphs or edge-noise settings?

---

### Official Review · Reviewer_PJey · 2025-10-30

**Soundness:** 1
**Presentation:** 2
**Contribution:** 1
**Rating:** 2
**Confidence:** 5

**Summary:**

This paper proposes an unsupervised graph denoising method, GraphDenoiser, which trains the noise prediction model through noise data sampling and iterative training. It claims to outperform multiple baselines on symmetric/asymmetric noise-injected graph datasets.

**Strengths:**

- The paper discusses an important problem: how to handle noisy data in graph data.

**Weaknesses:**

- Effectiveness of the Method is Doubtful: a) The noise distribution sampled according to the strategy in the paper likely differs significantly from real noise distributions, which raises doubts about the utility of the denoising model. 2) The noise predictor uses an MLP structure, which is likely unsuitable for graph noise structures.

- Experiments use Self-Sampled Noise: The experiments rely on self-sampled noise and do not test with real noisy datasets, such as NoisyGL or BeGIN.

- Lack of Generalization Experiments: There is no discussion of the generalization performance of GraphDenoiser.

- Writing Issues: The paper's writing (including figures) is unclear, and the formalization of equations could be improved.

**Questions:**

- Please explain the specific structure of the AutoEncoder used.
- Please see the remaining questions in "Weakness".

---

### Note · Authors · 2025-11-13

I have read and agree with the venue's withdrawal policy on behalf of myself and my co-authors.